# Mesh-Free Unsupervised Learning-Based PDE Solver of Forward and Inverse problems

## Abstract

We introduce a novel neural network-based partial differential equations solver for forward and inverse problems. The solver is grid free, mesh free and shape free, and the solution is approximated by a neural network. We employ an unsupervised approach such that the input to the network is a points set in an arbitrary domain, and the output is the set of the corresponding function values. The network is trained to minimize deviations of the learned function from the *strong* PDE solution and satisfy the boundary conditions. The resulting solution in turn is an explicit smooth differentiable function with a known analytical form.

Unlike other numerical methods such as finite differences and finite elements, the derivatives of the desired function can be analytically calculated to any order. This framework therefore, enables the solution of high order non-linear PDEs. The proposed algorithm is a unified formulation of both forward and inverse problems where the optimized loss function consists of few elements: fidelity terms of $L_2$ and $L_\infty$ norms that unlike previous methods promote a strong solution. Robust boundary conditions constraints and additional regularizers are included as well. This setting is flexible in the sense that regularizers can be tailored to specific problems. We demonstrate our method on several free shape 2D second order systems with application to Electrical Impedance Tomography (EIT), diffusion and wave equations.

## 1 Introduction

Partial differential equations are fundamental in science and mathematics with wide applications in medical imaging, signal processing, computer vision, remote sensing, electromagnetism, economics and more. Classical methods such as finite differences, finite volume and finite elements are numerical discretization-based methods where the domain is divided into a uniform grid or polygon mesh. The differential equation is then reduced to a system of algebraic equations. These methods may have some limitations: the solution is numeric and may suffer from high condition number, highly dependent on the discretization and even the second derivative is sensitive to noise, see for example Garcia (2017); LeVeque (2007); Thomas (1995); Reddy (2005) and references therein.

In the last few years, deep learning and neural network-based algorithms are extensively used in pattern recognition, image processing, computer vision and more. Recently, the deep learning approach had been adopted to the field of PDEs as well by converting the problem into a machine learning one. In *Supervised learning*, the network maps an input to an output based on example input-output pairs. This strategy is used in inverse problems, where the input to the network is a set of observations/measurements (e.g. x-ray tomography, ultrasound) and the output is the set of parameters of interest (tissue density etc.) Feigin et al. (2018); Lucas et al. (2018); McCann et al. (2017); Seo et al. (2019). *Unsupervised learning* on the other hand is a self-learning mechanism where the natural structure present within a set of data points is inferred.

Algorithms for forward and inverse problems in partial differential equations via unsupervised learning were recently introduced. The *indirect* approach utilizes a neural network as a component in the solution. Li et al. (2018) for example, proposed the NETT (Network Tikhonov) approach to inverse problems. NETT considers regularized solutions having a small value of a regularizer defined by a trained neural network. Khoo & Ying (2018) introduced a novel neural network architecture, Switch-Net, for solving the wave equation based inverse scattering problems via providing maps between

the scatterers and the scattered field. Han et al. (2018) developed a deep learning-based approach that can handle general high-dimensional parabolic PDEs. To this end, the PDEs are reformulated using backward stochastic differential equations. The latter is solved by a temporal discretization and the gradient of the unknown solution at each time step is approximated by a neural network. Li et al. (2019) approximate the inverse solution map of linear and nonlinear problems directly by a deep network.

*Direct* algorithms solve the forward and inverse problem PDEs by directly approximating the solution with a deep neural network. The network parameters are determined by the optimization of a cost function such that the optimal solution satisfies the PDE, boundary conditions and initial conditions. Chiaramonte & Kiener (2017) addressed the forward problem by constructing a one layer network which satisfies the PDE within the domain. The boundary conditions were analytically integrated in the cost function. They demonstrated their algorithm on the Laplace and hyperbolic conservation law PDEs. Sirignano & Spiliopoulos (2017) proposed a deep learning forward problem solver for high dimensional PDEs. Their algorithm was demonstrated on the American option free-boundary equation. Raissi et al. (2017) focused on continuous time models and solved the Burgers and Schrödinger equations, and Xu & Darve (2019) introduced a novel direct method for the inverse problem and demonstrated their algorithm on various PDEs.

In this work we address the forward and inverse PDE problems via a direct unsupervised method. Our key contributions are four fold: (1) inverse problems can be solved in the same framework as the forward problems. (2) In the both forward and inverse parts we extend the standard $L_2$-based fidelity term in the cost function by adding $L_\infty$-like norm. Moreover, (3) some regularization terms which impose a-priori knowledge on the solution can be easily incorporated. (4) Our construction exemplifies the ability to handle free-form domain in a mesh free manner.

Point (1) is essential for full tomography-like solutions as will be explained in the sequel. The extension of the loss function by the $L_\infty$-like norm is fundamental. This term promotes a strong solution of the PDE. The $L_2$ term, used in recent studies, aims only for weak solutions of the PDE. Weak solutions may differ from the strong solutions by a set of isolated points where the function is not continuous. This is not a merely theoretical issue but strongly affects the quality of the result as we empirically demonstrate. In unsupervised learning of ill-posed problems regularization is crucial. Choosing the right regularizer and the ability to incorporate it in the formulation is of prime importance. Our formalism integrates such regularizations in a natural way. We demonstrate our algorithm by a second order elliptic equation, in particular the Electrical Impedance Tomography (EIT) application on a circular and three other arbitrary domains. We additionally solve the inverse problem of diffusion and wave equations.

## 2 MATHEMATICAL FORMULATION

Let $\Omega$ be a bounded open and connected subset of $\mathbb{R}^d$ where $d$ is the dimension. A differential operator $\mathcal{L}$ acting on a function $u(\mathbf{x}) : \mathbb{R}^d \to \mathbb{R}$ is defined as

$$\mathcal{L}u(\mathbf{x}) := \Big(a_n(\mathbf{x}) \cdot \mathcal{D}^n + a_{n-1}(\mathbf{x}) \cdot \mathcal{D}^{n-1} + \ldots + a_1(\mathbf{x}) \cdot \mathcal{D} + a_0(\mathbf{x})\Big)u(\mathbf{x}), \quad (1)$$

where $\mathcal{D}^n$ is the nth order d-dimensional derivative and $a_0(\mathbf{x}), \ldots, a_n(\mathbf{x})$ are the coefficients. Consider the partial differential problem with Dirichlet boundary conditions

$$\begin{aligned} \mathcal{L}u(\mathbf{x}) &= f(\mathbf{x}), \quad \mathbf{x} \in \Omega \subset \mathbb{R}^d \\ u(\mathbf{x}) &= u_0(\mathbf{x}), \quad \mathbf{x} \in \partial\Omega, \end{aligned} \quad (2)$$

where $f(\mathbf{x}) : \mathbb{R}^d \to \mathbb{R}$ is a given function. The *forward problem* solves $u(\mathbf{x})$ given the coefficients $\theta := \{a_0(\mathbf{x}), \ldots, a_n(\mathbf{x})\}$ while the *inverse problem* determines the coefficients set $\theta$ given $u(\mathbf{x})$.

The proposed algorithm approximates the solutions in both problems by neural networks $u(\mathbf{x}; w_u)$, $\{a_j(\mathbf{x}; w_{a_j})\}$ such that the networks are parameterized by $w_u, \{w_{a_j}\}$, and the input to the network is $\mathbf{x} \in \mathbb{R}^d$. Figure 1 depicts a network architecture of $u$ in $\mathbb{R}^2$. The network consists of few fully connected layers with *tanh* activation and linear sum in the last layer. The network is trained to satisfy the PDE with boundary conditions by minimizing a cost function. In the forward problem

$$\mathcal{F}(u) = \lambda\|\mathcal{L}u - f\|_2^2 + \mu\|\mathcal{L}u - f\|_\infty + \|u - u_0\|_{1,\partial\Omega} + \mathcal{R}^F(u), \quad (3)$$

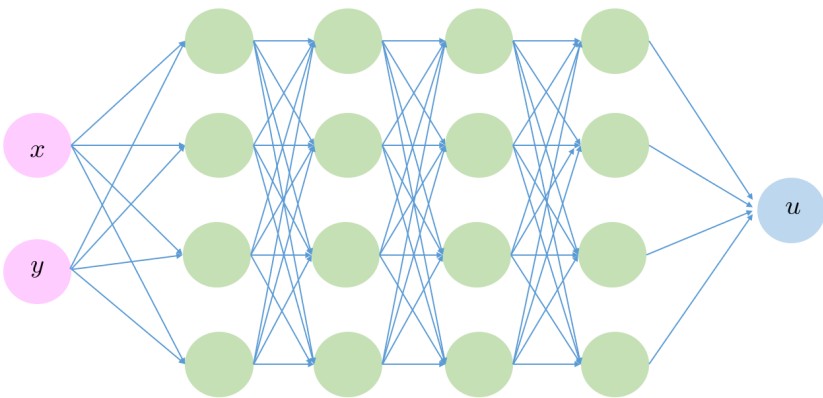

Figure 1: Network architecture: the point $(x, y) \in \mathbb{R}^2$ serves as an input and the value $u(x, y)$ as the output.

and in the inverse problem

$$\mathcal{I}(\theta) = \lambda\|\mathcal{L}u - f\|_2^2 + \mu\|\mathcal{L}u - f\|_\infty + \|\theta - \theta_0\|_{1,\partial\Omega} + \mathcal{R}^I(\theta). \tag{4}$$

The first two terms enforce the solution to satisfy the equation. The first term minimizes the error in the $L_2$ sense while the second term minimizes the maximal error. Ideally, the solution to the PDE has to be satisfied at every point $\mathbf{x}$. This is considered as a *strong* solution. In the deep learning framework the optimization is carried out on mini batches of points. The first term minimizes the equation in the integral or average sense and is therefore insensitive to a point jump in the value of the integrand. It therefore yields a *weak* solution. This manifests itself by point discontinuities of $u(\mathbf{x})$ that are empirically seen as well. The $L_\infty$ term promotes a strong solution since it handles possible discontinuous points where the value of the week solution is different than the strong one. We consider this discrepancy as an error. Practically, we use a relaxed version of the $L_\infty$ norm where we take the top-$K$ errors. Then, at every mini batch the training procedure minimizes the equation at $K$ different points having the largest errors. The third term imposes boundary conditions where $u_0$ and $\theta_0$ are the boundary values of $u(\mathbf{x})$ and $\theta(\mathbf{x})$. We used the $L_1$ norm on the boundary so that the solution is robust to measurements noise. The last term is a regularizer which can be tailored to the application. There are few advantages of this setting. First, the solution is a network and therefore an explicit function. It is a smooth analytic function and is therefore *analytically differentiable*. It was recently proven that for such functions, e.g. Lipchitz with constant $\nu$, the approximation of the function by a fully connected network $\Gamma$ is given by

$$\|u(x) - \Gamma\|_{L^p([0,1]^d)} \leq 40\nu\sqrt{d}N^{-2/d}L^{-2/d},$$

where $p \in [0, \infty)$, $N$ number of neurons in a layer, and $L$ number of layers Shen et al. (2019). In addition, the proposed framework enables setting a prior physical knowledge on the solution by designing the regularizers $\mathcal{R}^F$ and $\mathcal{R}^I$. Moreover, the training procedure is mesh free. Unlike finite differences or finite elements methods, we use *random* points in the domain and its boundary in the course of the optimization of equation 3 and equation 4, see Figure 2. This means that the solution does not depend upon a coordinate mesh and we can also define an arbitrary regular domain $\Omega$.

## 3 APPLICATION TO ELECTRICAL IMPEDANCE TOMOGRAPHY

Consider the following elliptic equation which is a special case of equation 1,

$$\begin{aligned} \nabla \cdot \Big(\sigma(\mathbf{x})\nabla u(\mathbf{x})\Big) = 0, \quad & \mathbf{x} \in \Omega \subset \mathbb{R}^2 \\ u(\mathbf{x}) = u_0(\mathbf{x}), \quad & \mathbf{x} \in \partial\Omega, \end{aligned} \tag{5}$$

where $\nabla := (\partial/\partial_x, \partial/\partial_y)$. We assume that $0 < \sigma(\mathbf{x}) \in C^1(\Omega)$, which guarantees existence and uniqueness of a solution $u \in C^2(\Omega)$ Evans (2010).

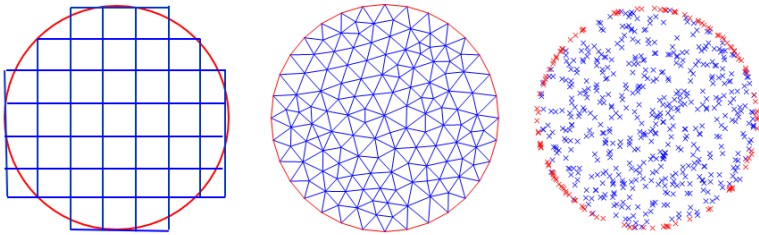

Figure 2: Left to right: finite differences grid, finite elements mesh and random points samples used in the proposed algorithm.

The elliptical system equation 5 was addressed by Siltanen et al. (2000) in the context of Electrical Impedance Tomography (EIT) which is a reconstruction method for the inverse conductivity problem. The function $\sigma$ indicates the electrical conductivity density, and $u$ is the electrical potential. An electrical current

$$\psi_{n,\varphi} = \sigma \frac{\partial u_n}{\partial \nu}\Big|_{\partial\Omega} = \frac{1}{\sqrt{2\pi}} \cos(n\kappa + \varphi), \;\; n \in \mathbb{Z} \tag{6}$$

is applied on electrodes on the surface $\partial\Omega$, where $\kappa$ is the angle in polar coordinate system along the domain boundary, $n$ is the current frequency, $\varphi$ is the phase and $\nu$ is the normal unit. The resulting voltage $u|_{\partial\Omega} = u_0$ is measured through the electrodes. The conductivity $\sigma$ is determined from the knowledge of the Dirichlet-to-Neumann map or voltage-to-current map

$$\Lambda_\gamma : u|_{\partial\Omega} \to \sigma \frac{\partial u_n}{\partial \nu}\Big|_{\partial_\Omega},$$

Mueller & Siltanen (2012); Alsaker & Mueller (2018); Fan & Ying (2019).

We demonstrate our framework by solving the forward and inverse problem of equation 5 which is a first step towards a full tomography. Following Mueller & Siltanen (2012), we simulate the voltage measurement $u|_{\partial\Omega}$ by the Finite Elements Method (FEM) given three variants of a conductivity phantom $\sigma(\mathbf{x})$ depicted in Figure 3. We calculate the FEM solution with different triangle mesh densities and select as ground truth the one such that finer meshes do not improve the numerical solution. With our suggested method, the forward problem determines the electrical potential $u$ in the whole domain $\Omega$ given $\sigma$, while the inverse problem uses the approximated $u$ and calculates the conductivity $\sigma$ given that $\sigma|_{\partial\Omega} = \sigma_0$.

## 4 FORWARD PROBLEM

In the forward problem the conductivity $\sigma(x_i)$ and boundary conditions $u_0(x_b)$ are given for random points set $\{x_i\} \in \Omega \subset \mathbb{R}^2$, $\{x_b\} \in \partial\Omega \subset \mathbb{R}^2$ with sets size of $N_s$ and $N_b$ respectively. A neural network having the architecture shown in Figure 1 approximates $u(\mathbf{x})$. Let

$$\mathcal{L}_i := \nabla \cdot \Big(\sigma(x_i)\nabla u(x_i)\Big). \tag{7}$$

The cost function equation 3 is then rewritten as

$$\mathcal{F}\Big(u(\mathbf{x}; w_u)\Big) = \frac{\lambda}{N_s} \sum_{i=1}^{N_s} |\mathcal{L}_i|^2 + \frac{\mu}{K} \sum_{k \in \text{top}_K(|\mathcal{L}_i|)} |\mathcal{L}_k| + \frac{1}{N_b} \sum_{b=1}^{N_b} \Big|u(x_b) - u_0(x_b)\Big| + \alpha\|w_u\|_2^2. \tag{8}$$

The first term is the $L_2$ norm of the differential operator, the second term is a relaxed version of the infinity norm where we take the mean value of the top-K values of $|\mathcal{L}_i|$. There is some balance between small $K$ which accounts for few error points and can have negligible effect in the training procedure, and a large $K$ which resembles the effect of the $L_2$ norm. In all our experiments we empirically selected $K = 40$ as an intermediate value. The third term imposes the boundary conditions and the last term serves as a regularizer of the network parameters.

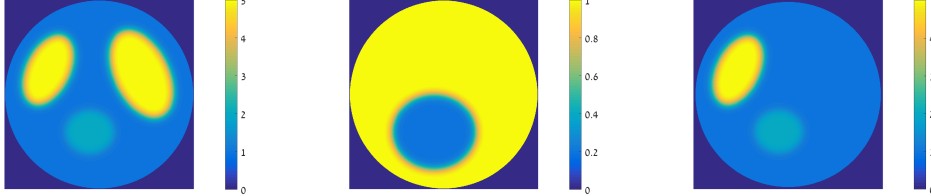

Figure 3: Synthetic conductivities $\sigma(\mathbf{x})$ referred to as phantoms. Left to right: phantom 1, phantom 2 and phantom 3

Table 1: Forward problem results for phantom 1 compared with the DGM method  Sirignano & Spiliopoulos (2017)

| | n | $\varphi$ | $u$: DGM | | $u$: Proposed | | $u_x$: DGM | | $u_x$: Proposed | |
|---|---|---|---|---|---|---|---|---|---|---|
| | | | MSE | PSNR | MSE | PSNR | MSE | PSNR | MSE | PSNR |
| 1 | 1 | 0 | 2.50e-1 | 9.02 | **2.38e-4** | **39.23** | 6.01e-6 | 11.17 | **7.11e-9** | **40.43** |
| 2 | 1 | $\pi/8$ | 4.20e-2 | 14.90 | **1.13e-4** | **40.60** | 7.95e-7 | 19.27 | **5.92e-9** | **40.55** |
| 3 | 1 | $\pi/4$ | 5.46e-2 | 13.54 | **1.23e-4** | **40.03** | 1.87e-6 | 13.36 | **5.62e-9** | **38.58** |
| 4 | 2 | 0 | 5.12e-2 | 16.90 | **4.50e-5** | **47.46** | 1.64e-6 | 14.28 | **2.81e-9** | **41.94** |
| 5 | 2 | $\pi/4$ | 6.11e-2 | 6.51 | **8.31e-5** | **35.18** | 1.99e-6 | 11.08 | **2.74e-9** | **39.69** |

The first phantom is shown in Figure 3 left. The impedance values associated with the background ellipses and circle were 1, 5 and 2 respectively. The original piecewise constant function $\sigma$ was slightly smoothed by a Gaussian kernel.

Figure 10 in the Appendix shows the forward problem results for current $\psi$ with $n = 1$ and $\varphi = \pi/8$. The left column is the FEM solution which is referred to as ground truth, where the top row indicates the solution $u(\mathbf{x})$ and the bottom row the derivative of $u(\mathbf{x})$ with respect to the first coordinate $x$ calculated as the finite difference approximation of the FEM result. The middle column depicts the outcome of the proposed method where $\partial u/\partial x$ is an analytical derivative of our result. The right column shows the outcome of the DGM method Sirignano & Spiliopoulos (2017) which is a special case of  equation 3 with $\lambda = 1$, $\mu = 0$, $L_2$ norm of the boundary conditions constraint with no regularizers. Quantitative results of the mean square error (MSE) and peak signal-to-noise ratio (PSNR) are summarized in Table 1. Clearly, the proposed method outperforms the DGM method since the weighting parameters, the $L_\infty$ norm and the network weights regularization play a significant role in the loss function.

Table 2: Forward problem results of phantom 2 given a circular domain $\Omega$, and phantom 3 with domains $\Omega_1$, $\Omega_2$ and $\Omega_3$ as defined in Figure 5

| | phantom | n | $\varphi$ | $u$ | | $u_x$ | |
|---|---|---|---|---|---|---|---|
| | | | | MSE | PSNR | MSE | PSNR |
| 1 | $2, \Omega$ | 1 | 0 | 2.86e-4 | 47.43 | 1.70e-8 | 40.61 |
| 2 | $2, \Omega$ | 1 | $\pi/2$ | 1.74e-3 | 38.90 | 1.03e-8 | 33.79 |
| 3 | $2, \Omega$ | 2 | 0 | 1.29e-4 | 45.52 | 3.26e-9 | 41.18 |
| 4 | $2, \Omega$ | 2 | $\pi/2$ | 1.30e-4 | 45.49 | 8.63e-4 | 37.07 |
| 5 | $3, \Omega_1$ | 1 | $\pi/4$ | 6.42e-5 | 47.16 | 5.64e-9 | 39.43 |
| 6 | $3, \Omega_1$ | 2 | $\pi/4$ | 1.08e-4 | 34.03 | 2.61e-9 | 41.32 |
| 7 | $3, \Omega_2$ | 1 | $\pi/4$ | 1.08e-4 | 44.91 | 4.51e-9 | 40.39 |
| 8 | $3, \Omega_2$ | 2 | $\pi/4$ | 5.93e-5 | 36.64 | 2.74e-9 | 41.11 |
| 9 | $3, \Omega_3$ | 1 | $\pi/4$ | 1.22e-4 | 44.37 | 9.17e-9 | 37.31 |
| 10 | $3, \Omega_3$ | 2 | $\pi/4$ | 1.74e-4 | 31.96 | 2.24e-9 | 41.99 |

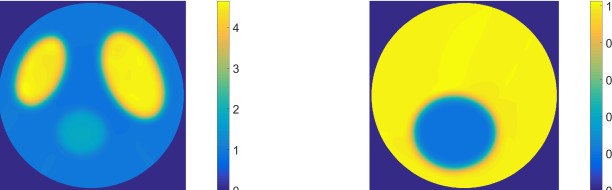

Figure 4: Reconstructed conductivity $\sigma(\mathbf{x})$ using the generalized inverse problem. Left: phantom 1, Right: phantom 2. MSE and PSNR are summed up in rows 1-2 of Table 3.

Table 3: Reconstruction of $\sigma(\mathbf{x})$ using the generalized inverse problem

|   | phantom | MSE | PSNR |
|---|---------|------|-------|
| 1 | $1, \Omega$ | 4.06e-2 | 27.90 |
| 2 | $2, \Omega$ | 9.04e-5 | 40.44 |
| 3 | $3, \Omega_1$ | 1.07e-2 | 33.67 |
| 4 | $3, \Omega_2$ | 5.9e-3 | 36.27 |
| 5 | $3, \Omega_3$ | 1.12e-2 | 33.51 |

The forward problem was repeated using phantom 2 where the background and circle conductivities were 1 and 0.2 respectively (Figure 3, middle). Four different current combinations were applied. Quantitative results are summarized in rows 1-4 of Table 2. Figure 11 in the Appendix shows the results for both $u$ and $\partial u/\partial x$ for $n = 2$ and $\varphi = \pi/2$. The right column presents the relative error defined as $e(x,y) = (u_{\text{fem}}(x,y) - u(x,y))/\max(u_{\text{fem}})$.

## 5   INVERSE PROBLEM FOR EIT

In the inverse problem, the electrical potential $u(\mathbf{x})$ is known while $\sigma(\mathbf{x})$ is unknown. Since we have a network which approximates $u(\mathbf{x})$, we can evaluate it at any point $\mathbf{x}$. The objective function equation 4 then takes the form

$$
\mathcal{I}\Big(\sigma(\mathbf{x}; w_\sigma)\Big) = \frac{\lambda}{N_s} \sum_{i=1}^{N_s} |\mathcal{L}_i|^2 + \frac{\mu}{K} \sum_{k \in \text{top}_K(|\mathcal{L}_i|)} |\mathcal{L}_k|
$$
$$
+ \frac{1}{N_b} \sum_{b=1}^{N_b} \Big| \sigma(x_b) - \sigma_0(x_b) \Big| + \alpha \|w_\sigma\|_2^2 + \frac{\beta}{N_s} \sum_{i=1}^{N_s} |\nabla \sigma(x_i)|^p.
$$
(9)

As in the forward problem, the first two terms enforce $\sigma$ to satisfy the PDE, where $\mathcal{L}_i$ is defined in equation 7. The third term imposes the boundary conditions, and the fourth regularizes the network parameters. A physical a priori knowledge regarding $\sigma$ is exploited in this application. The conductivity is assumed to have well defined sub-regions. We therefore design the fifth term to have sparse edges via the total variation regularization ($p = 1$).

Additional inverse problem generalization may exploit multiple $u$ approximations for several currents $\psi_j$. The $\sigma$ calculation thus, simultaneously relies on all $\{u_j\}$, resulting a more stable solution. Let

$$
\mathcal{L}_{ij} := \nabla \cdot \Big( \sigma(x_i) \nabla u_j(x_i) \Big).
$$
(10)

Then equation 9 is generalized to

$$
\mathcal{J}_j\big(\sigma(\mathbf{x}; w_\sigma)\big) = \frac{\lambda}{N_s} \sum_{i=1}^{N_s} |\mathcal{L}_{ij}|^2 + \frac{\mu}{K} \sum_{k \in \text{top}_K(|\mathcal{L}_{ij}|)} |\mathcal{L}_{kj}|,
$$
(11)

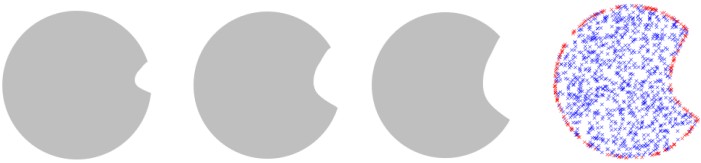

Figure 5: Free form shapes. Left to right: $\Omega_1$, $\Omega_2$, $\Omega_3$ and sample points of $\Omega_3$. The red points indicate the boundary.

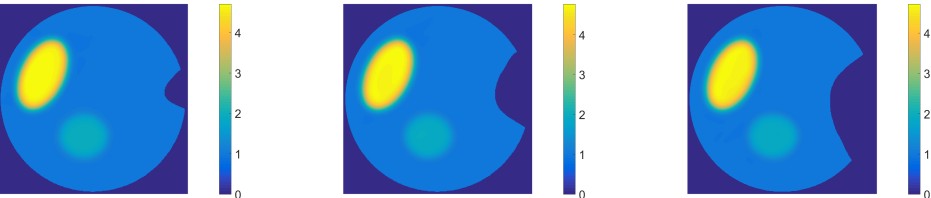

Figure 6: Reconstructed conductivity $\sigma(\mathbf{x})$ using the generalized inverse problem given phantom 3 applied on different domains. Left to right: $\Omega_1$, $\Omega_2$, $\Omega_3$. MSE and PSNR are given in rows 3-5 of Table 3

and

$$\mathcal{I}\Big(\sigma(\mathbf{x}; w_\sigma)\Big) = \sum_j \mathcal{J}_j\Big(\sigma(x; w_\sigma)\Big) + \frac{1}{N_b}\sum_{b=1}^{N_b}\Big|\sigma(x_b) - \sigma_0(x_b)\Big| + \alpha\|w_\sigma\|_2^2 + \frac{\beta}{N_s}\sum_{i=1}^{N_s}|\nabla\sigma(x_i)|^p.$$

(12)

Adequate reconstruction results by the generalized inverse problem are shown in Figure 4 and rows 1-2 in Table 3. In both phantoms we used four current combinations $\psi_{n,\varphi}$: $(1, 0)$, $(1, \pi/4)$, $(2, 0)$, $(2, \pi/4)$, and $\sigma_0 = 1$.

## 6 FREE SHAPE GEOMETRY

We applied the proposed method to arbitrary domains $\Omega_1$, $\Omega_2$ and $\Omega_3$, see Figure 5. The random sample points within the domain and along its boundary can be easily obtained as can be seen in Figure 5 right. Forward problem results applied on phantom 3 are presented in rows 5-10 of Table 2. Figure 12 in the Appendix shows the results for $n = 2$ and $\varphi = \pi/4$ for the three domains. The outcome of the generalized inverse problem equation 12 is shown in Figure 6 and rows 3-5 of Table 3.

## 7 ADDITIONAL INVERSE PROBLEM EXAMPLES

In this section we exemplify the proposed algorithm with additional inverse problem setting introduced by Xu & Darve (2019). Unknown parameters which are approximated by neural networks are inferred out of few measurements. The differential operators are approximated in finite differences schemes discretized by $\Delta t$ and $\Delta h$, the temporal and spatial grid spacings. In the two following examples we applied the cost function equation 4 with total variation and network parameters regularizations.

### 7.1 DIFFUSION EQUATION

Diffusion PDE is given by

$$\frac{\partial u(\mathbf{x}, t)}{\partial t} = C(\mathbf{x})\nabla^2 u(\mathbf{x}, t) + f(\mathbf{x}, t) \quad (\mathbf{x}, t) \in [-1, 1]^2 \times [0, T],$$

where $t$ denotes the time and the conductivity $C(\mathbf{x})$ is the unknown parameter. In the following we solve the equation in 2D where $\mathbf{x} := (x, y)$. Given two measurements at adjacent times $u(x, y, t_1)$ and $u(x, y, t_1 + \Delta t)$, the conductivity $C(x, y)$ is approximated as a neural network and is determined as the minimizer of equation 4 with

$$\mathcal{L}(u) - f = \frac{u(x, y, t_1 + \Delta t) - u(x, y, t_1)}{\Delta t} - C(x, y) \frac{D^2 u(x, y, t_1) + D^2 u(x, y, t_1 + \Delta t)}{2(\Delta h)^2} - f(x, y, t_1),$$

where $D^2$ is the central scheme approximation for the Laplacian on 3x3 stencil. The analytical 2D conductivity function is given by

$$C(x, y) = 1 + e^{-(12xy)^2},$$

and

$$u(x, y, t) = e^{-\pi^2 t} \sin \pi x \cos(\pi y + \pi/4),$$
$$f(x, y, t) = \pi^2 u(x, t)(2C(x, y) - 1).$$

The measurements were taken at $t_1 = 0.1$ with $\Delta t = 0.001$ and $\Delta h = 0.002$. The result is shown in Figure 7. As can be seen the smoothness regularizer and the $L_\infty$ terms increase the reconstruction accuracy. In addition, we added few Gaussian noise levels to the measurements. Quantitative results are summarized in rows $1 - 4$ of Table 4.



Figure 7: Reconstructed conductivity $C(x, y)$. Left: ground truth, Middle: proposed method, Right: Xu & Darve (2019). MSE and PSNR are given in rows 1-4 of Table 4

## 7.2 WAVE EQUATION

The second example is the wave equation given by

$$\frac{\partial^2 u(x, y, t)}{\partial t^2} = \eta^2(x, y) \nabla^2 u(x, y, t) \quad ((x, y), t) \in [-1, 1]^2 \times [0, T].$$

The unknown parameter is the velocity field $\eta(x, y)$. Its analytical value is given by

$$\eta(x, y) = 0.1 + e^{-((x-0.5)^2) + (y-0.1)^2)},$$

and the two initial conditions are

$$u(x, y, 0) = e^{-10x^2 - 3y^2},$$

and

$$u(x, y, \Delta t) = e^{-10(x-0.001)^2 - 3(y-0.001)^2}.$$

Solutions for all time steps are recursively calculated using the finite differences scheme using the correct $\eta(x, y)$. Given three consecutive values of $u(x, y, t)$ the velocity field $\eta(x, y)$ is extracted by the minimization of equation 4 where

$$\mathcal{L}(u) = \frac{u(x, y, t_1 + \Delta t) - 2u(x, y, t_1) + u(x, yt_1 - \Delta t)}{(\Delta t)^2} - \eta^2(x, y) \frac{D^2 u(x, y, t_1)}{(\Delta h)^2}.$$

Reconstruction result is depicted in Figure 8 showing the advantage of the proposed method. Additional quantitave results of noisy measurements are given in rows 5-8 of Table 4.

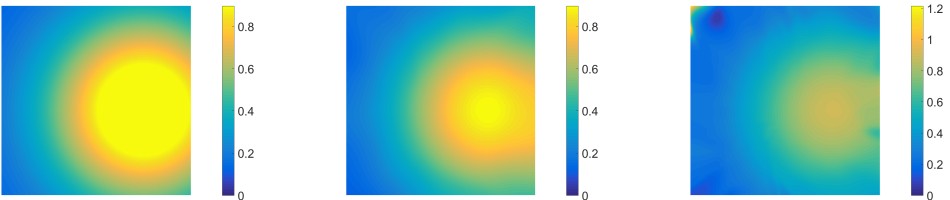

Figure 8: Reconstructed velocity field $\eta(x, y)$. Left: ground truth, Middle: proposed method, Right: Xu & Darve (2019). MSE and PSNR are given in rows 5-8 of Table 4

Table 4: Inverse problem results for diffusion and wave equations

|  | Equation | $\sigma_{noise}$ | Xu & Darve (2019) MSE | PSNR | Proposed MSE | PSNR |
|---|---|---|---|---|---|---|
| 1 | Diffusion | 0 | 4.90e-4 | 39.12 | **2.04e-4** | **42.92** |
| 2 | Diffusion | 1e-7 | 0.259 | 11.89 | **0.249** | **12.06** |
| 3 | Diffusion | 5e-7 | 0.335 | 10.77 | **0.316** | **11.02** |
| 4 | Diffusion | 1e-6 | 0.448 | 9.50 | **0.438** | **9.60** |
| 5 | Wave | 0 | 0.015 | 19.01 | **0.013** | **19.53** |
| 6 | Wave | 3e-9 | 0.049 | 13.92 | **0.015** | **19.16** |
| 7 | Wave | 3.5e-9 | 0.056 | 13.34 | **0.018** | **18.32** |
| 8 | Wave | 4e-9 | 0.059 | 13.11 | **0.029** | **16.19** |

## 8 IMPLEMENTATION DETAILS

The network architecture had 4 layers having 26, 26, 26 and 10 neurons. The algorithm was implemented by TensorFlow Ten (2015) using the ADAM optimizer which is a variant of the SGD algorithm. We used almost the same hyper parameters set in our experiments. Batch size=1000, decaying learning rate starting at $5e - 3$. The learning rate was factored by 0.8 every 200 epochs, $N_s = 80000$, $N_b = 1200$, $\lambda = 0.01$, $\alpha = 1e - 8$, $K = 40$, $\mu = 0.01$, and $\beta = 0.01$. In the diffusion and wave equation case we set $\lambda = 1$, $\beta = 1e - 4$, $\mu = 1e - 4$ and $\alpha = 1e - 5$. Running time, for the EIT, on Intel i7-8650u CPU was about 15 minutes for the forward problem and 13 minutes for the generalized inverse problem. For the diffusion and wave equation the running time was about 5 minutes.

Finally, we performed a sensitivity analysis of the algorithm with respect to the number of sample points $N_s$. Figure 9 shows the sensitivity of the forward problem algorithm. As expected, with large number of points we obtain a plateau while with small and intermediate number (up to 50000) the error is not monotonic due to local minima.

## 9 DISCUSSION

Deep networks by their nature use compositions of simple functions such as matrix multiplication and non-linear activations like sigmoid or tanh. This structure (i) enables the approximation of an arbitrary function Hornik (1989) and (ii) is inherently differentiable. The network architecture dictates the number of degrees of freedom which in turn enables the expressibility of complex functions. In this work we present a unified framework for the solution of forward and inverse problems in partial differential equations in an arbitrary domain. The algorithm relies on direct approximation of the unknown function by a neural network which yields an *analytical* smooth solution. The network is trained to satisfy the PDE and boundary conditions in an unsupervised fashion by the minimization of a cost function. The optimization procedure depends on random points set within the domain and its boundary. The problem is therefore mesh free with free-form domain.

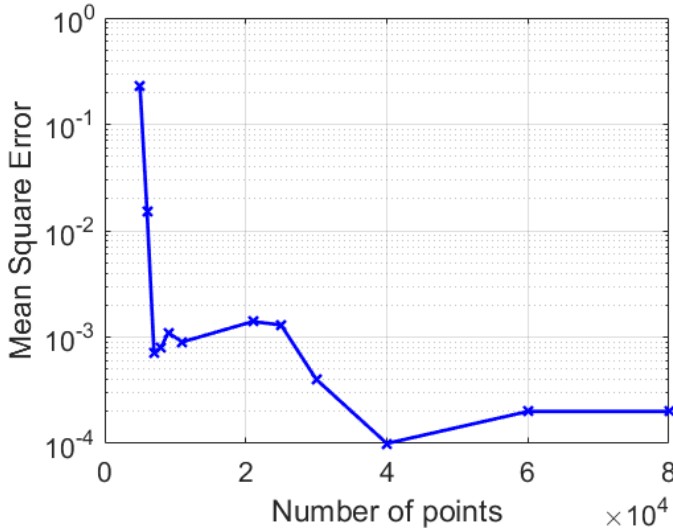

Figure 9: Sensitivity analysis of the algorithm with respect to the number of sample points.

While recent works formulate the cost function in both forward and inverse problems in the $L_2$ sense, we introduce a broadened framework. The relaxed $L_\infty$ fidelity term was designed to approximate a strong solution and control possible discrepancies between the weak and the strong solutions. The $L_1$ norm on the boundary conditions is more robust to measurement noise than the $L_2$ norm. The additional regularizer (total variation in our examples) has a significant role in the optimization process where known physical properties like smoothness are imposed on the solution. Lastly, the network regularizer controls the network weights and prevents over fitting. These additional terms make the proposed algorithm more accurate and robust compared to other learning-based methods. We also stress the robustness of our approach exemplified by having almost the same hyper parameters set in our experiments.

This framework alleviates several problems of the finite differences and finite elements methods. In particular meshing, discretization and derivatives approximation are solved in a simple and natural way. Numerical solutions of PDEs in an arbitrary domain are of extreme importance, in particular in medical imaging applications. The framework is demonstrated by an elliptic system in $\mathbb{R}^2$ applied to Electrical Impedance Tomography for both forward and inverse problems, diffusion and wave equations. Promising results were achieved for complex and non monotonic functions. Rigorous analysis of the approximation error and its relation to the network architecture and design are under current study. Future research includes also full tomography solution, higher dimensional problems and other classes of PDEs such as dynamic non-linear equations (Burgers, Navier-Stokes etc.)

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

## A  APPENDIX

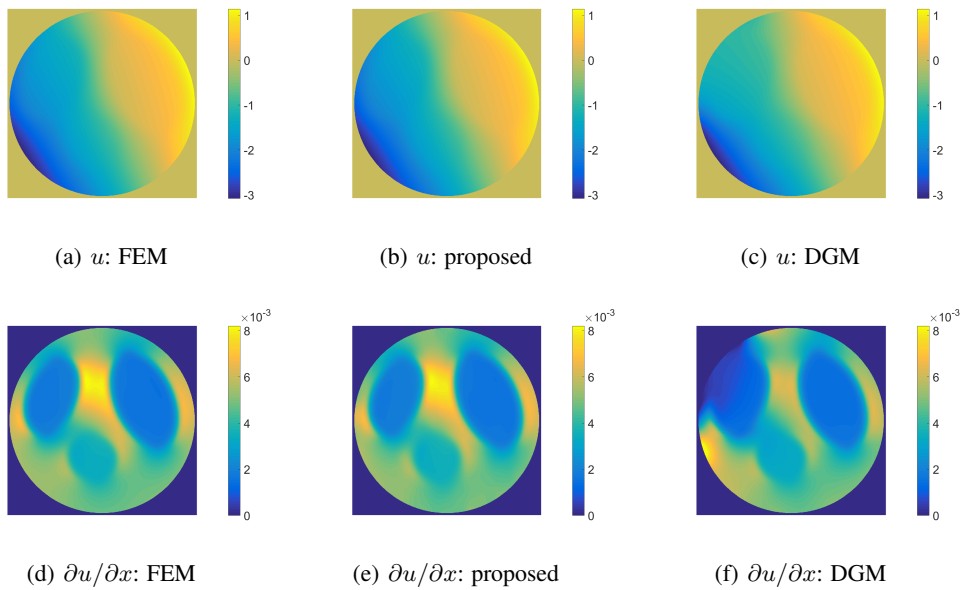

(a) $u$: FEM

(b) $u$: proposed

(c) $u$: DGM

(d) $\partial u/\partial x$: FEM

(e) $\partial u/\partial x$: proposed

(f) $\partial u/\partial x$: DGM

Figure 10: Forward problem results of $u(\mathbf{x})$ and $\partial u(\mathbf{x})/\partial x$ for current frequency $n = 1$ and phase $\varphi = \pi/8$ given phantom 1. Left column: ground truth (FEM). Middle column: proposed method. Right column: DGM method. MSE and PSNR are reported in row 2 of Table 1

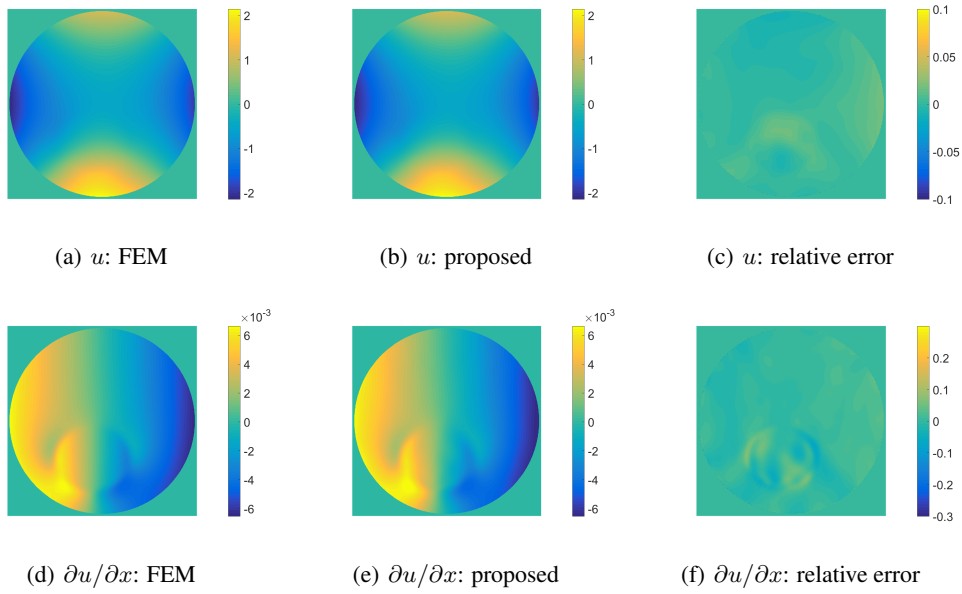

(a) $u$: FEM  (b) $u$: proposed  (c) $u$: relative error

(d) $\partial u/\partial x$: FEM  (e) $\partial u/\partial x$: proposed  (f) $\partial u/\partial x$: relative error

Figure 11: Forward problem results of $u(\mathbf{x})$ and $\partial u(\mathbf{x})/\partial x$ for current frequency $n = 2$ and phase $\varphi = \pi/2$ given phantom 2. Left column: ground truth (FEM). Middle column: proposed method. Right column: relative error. MSE and PSNR are reported in row 4 of Table 2

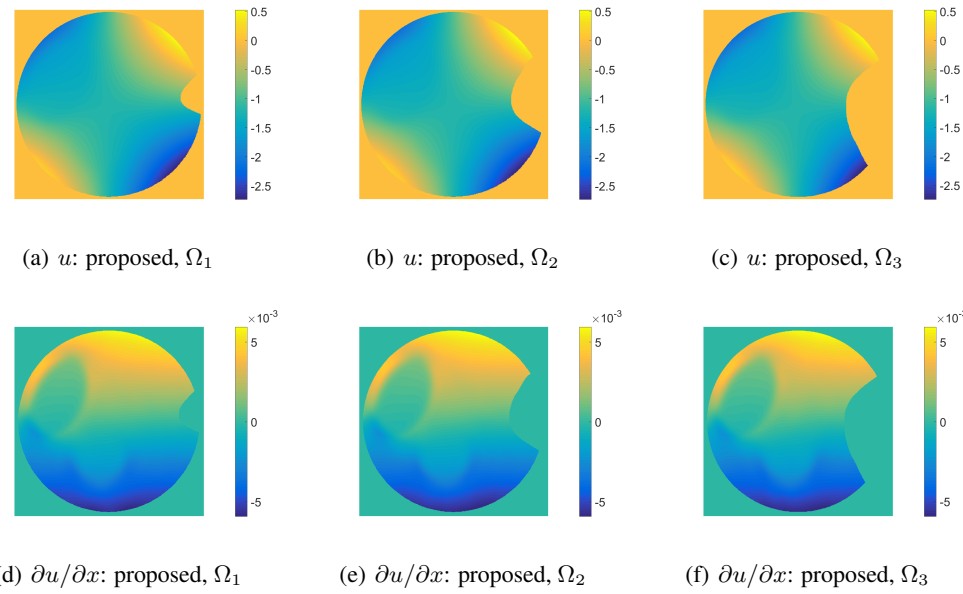

(a) $u$: proposed, $\Omega_1$  (b) $u$: proposed, $\Omega_2$  (c) $u$: proposed, $\Omega_3$

(d) $\partial u/\partial x$: proposed, $\Omega_1$  (e) $\partial u/\partial x$: proposed, $\Omega_2$  (f) $\partial u/\partial x$: proposed, $\Omega_3$

Figure 12: Forward problem results of $u(\mathbf{x})$ and $\partial u(\mathbf{x})/\partial x$ for current frequency $n = 2$ and phase $\varphi = \pi/4$ given phantom 3 applied on different domains. Left to right: $\Omega_1$, $\Omega_2$, $\Omega_3$. MSE and PSNR are reported in rows 6, 8 and 10 of Table 2.

