# OpenReview forum: "Mesh-Free Unsupervised Learning-Based PDE Solver of Forward and Inverse problems"
_ICLR.cc/2020/Conference — Reject_

### Official Review · AnonReviewer1 · 2019-10-18
**Official Blind Review #1**

**Rating:** 6

**Review:**

In this work, the authors propose an approach to solve forward and inverse problems in partial differential equations (PDEs) using neural networks. In particular, they consider a specific type of second-order differential operator combined with Dirichlet boundary conditions, and suggest how neural networks with suitable training objectives can be used to solve both types of problems. For a specific elliptic system relevant for Electrical Impedance Tomography (EIT), they then present numerical results obtained with their method and compare those to high-resolution finite element solutions and previous work.

The paper addresses a very general and important problem with wide ranging applications. People have solved PDEs using spectral, finite difference, finite volume or finite element methods for decades and there is a huge body of literate on this subject. The neural network based approach proposed in this paper seems general and simple with encouraging experimental results. However, there are several important points missing in the current paper based on which I would recommend rejecting it. However, I would be willing to increase the score if these points were addressed in sufficient detail.

Major comments:

1) Choice of problems

There are many important problems in the literature that belong to the general type of PDE considered in this work (e.g. in electrostatics). To appreciate and understand the merits and limitations of the proposed approach better, it would be highly necessary to apply it to a wider range of problems, including ones with known analytical solution.

2) Convergence tests

Convergence tests are an important part that is missing in the current paper. In particular, how does the error in the neural-network solution decay with respect to the number of random points used to train the network? Plots comparing a suitable error norm versus the training dataset size would be very informative. Furthermore, what are the conditions, if any, to guarantee convergence to the exact solution? The authors mention in the last paragraph of the Discussion that a more rigorous analysis will be published elsewhere. However, as error analysis is such an integral part of the study, I think it needs to be addressed, at least to some extent, already in the current paper.

3) Related work

I don’t think the current work is put in sufficient contrast with existing work. Several related papers are mentioned in the introduction and the experimental section includes one other method (GDM method of Sirignano & Spiliopoulos) for comparison. However, a more thorough discussion on how the current approach complements existing literature on neural network based PDE solvers would be in order.

Minor comments:

i) There are quite a few typos and grammar mistakes in the paper that need to be fixed. To give a few examples:
‘the natural structure presents’ -> ‘the natural structure present’
‘Shrödinger’ -> ‘Schrödinger’
‘approximated by neural network’ -> ‘approximated by a neural network’
‘on circular and’  -> ‘on a circular and’
‘forces the equation’ -> ‘enforces the equation’
etc...

ii) Eq. (7): There is a parenthesis missing on the LHS.

iii) The authors show many plots in Figs 3-9 but don’t comment much on the results in the main text. A lot of the results presented could therefore go into an appendix. I would find it better to present fewer results/plots and discuss those in more detail.

iv) It would be good to define the Nabla operator in Eq. 5 for completeness.

v) Please put references to equations and papers in parentheses or, at least, separate them otherwise from the sentence, to avoid confusion. For example:
- the reference to ‘Evans (2010)’ below Eq. 5, or
- the list of dangling references before the last paragraph on page 3,
- the reference to Sirignano & Spiliopoulos in the caption of Table 1,
etc...

vi) Please expand the captions in Fig. 3 to make the figure meaningful as a stand-alone, without having to read the entire text to understand what 'phantom' means. The same applies to all other captions.

vii) Perhaps it would be clearer to use ‘row’ instead of ‘line’ when referring to the results in the table.

viii) In Sec. 7 ‘s’ in ‘Ns’ should be a subscript.

ix) Explain what the acronym PSNR means.

x) Please elaborate on why you take the top k values in Eq. (7). What happens if you take more/less?

xi) First paragraph in Discussion: I am not sure the robustness w.r.t. the hyperparameters is really so surprising given that you apply the solver to very similar problems. If you could show that the same hyperparameters worked on a completely different problem, that would be much more interesting.

xii) Please add references for the first paragraph of the introduction.


*********************************************************
I increased my rating based on the revisions.
*********************************************************

**Experience Assessment:**

I have published one or two papers in this area.

**Review Assessment: Checking Correctness Of Derivations And Theory:**

I assessed the sensibility of the derivations and theory.

**Review Assessment: Checking Correctness Of Experiments:**

I assessed the sensibility of the experiments.

**Review Assessment: Thoroughness In Paper Reading:**

I read the paper at least twice and used my best judgement in assessing the paper.

---

> ### Author Response · Authors · 2019-11-08
> **Answers to the reviewer's concerns**
>
> We thank the reviewer for the constructive and supportive comments.
>
>
> 1. Thanks. We added two inverse  problem examples:  diffusion and wave equations in the presence of additive Gaussian noise and compared our results to Xu and Darve (2019).
>
> 2. Thanks. We added a plot that demonstrates the performance as a function of the number of points in the training phase.  As expected, with large number of points we obtain a plateau while with small and intermediate number (up to 50000) the error is not monotonic due to local minima.
>
> 3. .We added an approximation result by Shen et al. that gives a bound on the representation of a function in terms of the number of layers and the number of neurons in each layer.  The problem of bounding the error of approximation by a network is notoriously open problem. In our case the problem is more complex since the network approximates the function indirectly via the PDE. This is under current study.
>
> 4. Related work will be added to the introduction.
>
> Minor concerns:
> We highly appreciate the reviewer for carefully reading the manuscript  and for providing these corrections. . It will be corrected shortly.

---

### Official Review · AnonReviewer3 · 2019-10-23
**Official Blind Review #3**

**Rating:** 3

**Review:**

This paper presents an unsupervised learning approach to solve forward and inverse problems represented by partial differential equations. A framework is proposed based on the minimisation of loss functions that enforce the boundary conditions of the PDE solution and promote its smoothness. The method is then applied to solve forward and inverse problems in electrical impedance tomography.

Flexible machine learning approaches to solving partial differential equations is a subject of ongoing research. While the paper presents an elegant solution folding forward and inverse problems into a single framework, the presentation is missing a few important details which difficult the assessment of the contribution and favour a rejection of the paper. The main issues are insufficient experimental comparisons and a lack of theoretical support for the method.

Major issues:

1. When compared to DGM (Sirignano & Spiliopoulos, 2017), for the forward problem, the method only differs in the form of the loss function, which is almost identical, with the exception of the additional L-infinity term and the optional user-defined regularisers. The argument for the inclusion of the L-infinity loss is its high sensitivity to outliers, enforcing a smooth solution. (a) Why PDE solutions learned using the original loss from DGM, which also yields continuous functions, should present outliers in the first place? Moreover, the possibility of adding a user-defined regularizer seems to be a relatively simple extension. (b) What should the theoretical or practical implications for the extra user-defined regularisation term be?

2. The loss function for the inverse problem, which seems to be one of the paper’s contributions, misses a dedicated discussion. An important detail in this loss is the third term, which enforces boundary conditions for the coefficients at the boundary of the domain. In Equation 2, however, the coefficients only affect the PDE through the Lu term over the domain, not its boundary. So what does \theta_0 mean in Equation 4 then?

3. The paper proposes a general framework, but experimental results are presented for only one specific problem, the electrical impedance tomography. The generalisability of the method to more complex problems, such as PDEs with time components and a high-dimensional spatial domain, cannot be inferred. Adding experimental comparisons on higher-dimensional domains would strengthen the paper.

4. Experiments only present comparisons to relevant state-of-the-art methods (DGM) in the forward problem. There are no comparisons against other methods for the inverse and the free-shape geometry problems. For example, have the authors considered the method in [A]?
[A] Xu, Kailai, and Eric Darve. "The neural network approach to inverse problems in differential equations." arXiv preprint arXiv:1901.07758 (2019).

Minor issues:

1. The background on PDEs is relatively short for a machine learning conference. (a) There lacks an explanation on what the operator \mathcal{L} means. (b) Equation 1 lacks an explicit use of “u(x)”, instead of simply “u”, causing confusion with the dependence of the coefficients on “x”. (c) The meaning of the index subscripts on the partial derivatives is also not made clear, especially if “u” could be interpreted as a vector-valued function for someone unfamiliar with PDEs. Replacing “some u” by “some u:R^d\to\R” would already help.

2. What does “n” mean in the electrical current equations in Sec. 3?

3. The derivative of a scalar “u(x)” with respect to a vector “x” should be a vector. So what are the plots in figures 4, 5 and 8 showing when referring to du/dx? Is that the magnitude of the vector or the partial derivative with respect to a single spatial component?

4. What does “PSNR” stand for?

5. Indirect citations in the text should be enclosed by brackets using something like the “\citep” command from the package “natbib”.

6. In Table 1, there is a typo: “GDM”->”DGM”.

7. The context contains a few minor grammatical issues that can be distracting at times, but should be solvable by revision.

**Experience Assessment:**

I have read many papers in this area.

**Review Assessment: Checking Correctness Of Derivations And Theory:**

I assessed the sensibility of the derivations and theory.

**Review Assessment: Checking Correctness Of Experiments:**

I assessed the sensibility of the experiments.

**Review Assessment: Thoroughness In Paper Reading:**

I read the paper at least twice and used my best judgement in assessing the paper.

---

> ### Author Response · Authors · 2019-11-08
> **Answers to the reviewer's concerns**
>
> We thank the reviewer for the constructive comments.
>
> 1. We added an explanation for the L_inft term in the text. The point here is that the loss function is defined in the weak sense via a L_2 norm since we average the loss function over the minibatch The integral is insensitive
> to a point jump in the value of the integrand.  This manifest itself by  point discontinuities in u(x) that are seen empirically as well. We added the L_infty terms to have a solution in the strong sense.  We showed that this term is empirically crucial as  well.
>
> While supervised learning use directly the data as prior, unsupervised methods that tackle ill-posed problems must use a prior to steer the solution to the right subspace of possible solutions. The exact structure of the loss function and the choice of prior is not, in our eyes,  a “simple addition” but a major modeling issue which can handle noisy measurement and compensate ill-posed cases.
>
> 2. A discussion on the inverse problem  will be added in the paper.
> Thanks. This point is now  clarified in the paper. In Eq. 3 we demonstrate the solution of the forward problem so the coefficient is known in the domain and on the boundary.  In eq. 4 the inverse problem is done and there we need to use the boundary information of the parameter.
>
> 3. Thanks. We added two inverse problem examples: diffusion and wave equations in the presence of additive Gaussian noise with comparison to a Xu and Darve(2019).
>
> 4. Thanks for referring us to this publication that evaded our search. We  added now  comparisons to this paper. We exemplified our method in two dimensions in the presence of additive Gaussian noise.
>
> Minor issues:
> We appreciate the reviewer for carefully reading the manuscript and for providing these corrections. . It will be corrected shortly.

---

### Official Review · AnonReviewer2 · 2019-10-24
**Official Blind Review #2**

**Rating:** 3

**Review:**

Summary: In the paper, the authors purpose to use neural networks to model both the function $u$ and the parameters and in a sense, unify the forward and inverse problems. The authors demonstrate the work on Electrical Impedance Tomography problem. The authors also purpose a new regularizer, the sum of L2 and L_inf norm of the differential operator.

Concerns: there have been plenty of works that use neural networks to model the function $u$ for forward problems and another bunch of works that use neural networks to model parameters to do inverse problems.

It is not clear to me if combining the two will really give us benefits, since we are still doing these two problems separately. If we are doing some alternating training, unifying them could be useful.

The mesh free part is less interesting in my opinion. Works using feed forward neural networks are mesh free in general. And when you try to use the solution, or to compare with some ground truth generated by traditional methods, usually we still need to make the solution discrete to use it.

The experiments in the paper is limited. It compares with only one work in the forward task, but no comparison in the inverse problem. It is hard to evaluate its performance.

The theory is also needed. It is not very clear why L2 + L_inf regulation terms will help us.
After all, in computer science, conference papers are considered as final publications. So a more extensive studied is expected. I would suggest submit this work for a workshop.

Decision: This work need further experiments and theoretical analyze. I suggest weakly reject this paper.

**Experience Assessment:**

I have read many papers in this area.

**Review Assessment: Checking Correctness Of Derivations And Theory:**

N/A

**Review Assessment: Checking Correctness Of Experiments:**

I assessed the sensibility of the experiments.

**Review Assessment: Thoroughness In Paper Reading:**

I read the paper thoroughly.

---

> ### Author Response · Authors · 2019-11-08
> **Answers to the reviewer's concerns**
>
> We thank the reviewer for the constructive remarks
>
> 1. We compared to the main publication that solved the forward problem.
> Reviewer 3 referred us to a novel work in inverse problems. We prepared two more inverse problem examples. Diffusion and wave equations. We compared our results to Xu and Darve (2019).  We solved the inverse problems in 2D with several Gaussian noise levels. The results and comparisons are reported in Table 4 and figures 9,10 in the revised paper.
>
> 2. We would like to add that the main challenge is to perform a full tomography from data that is defined only on the boundary for the function and the parameter sigma.  For this to be feasible a good forward and inverse algorithm should be available and in the same framework.  The full tomography is under current study and we will have publishable results in a couple of months.
>
> 3. Quote: "And when you try to use the solution,
> or to compare with some ground truth generated by traditional methods, usually we still need to make the solution discrete to use it. "
> Answer:
> We should have been clearer on that point. The network is continuous in the input . In order to compare with the ground truth generated by traditional method we need to compare it on the grid where the traditional method is defined. The network value than was calculated on these grid points. The function value can be evaluated at any arbitrary point x. It was not discretized. An explanation for this will be added to the text.
>
> 4. We added comparisons for an inverse solver as well. We address diffusion and wave equations in two dimensions in the presence of Gaussian noise.
>
> 5. We  extended the explanation of the L_infty term in the text. The point here is that the loss function is defined in the weak sense via a L_2 norm since we average the loss function over the minibatch. The integral is insensitive
> to a point jump in the value of the integrand.  This manifest itself by point discontinuities of  u(x) that are seen empirically as well. We added the L_infty terms to have a solution in the strong sense.

---

### Author Response · Authors · 2019-11-08
**General comment**

We thank the reviewers for their constructive comments.  We added more examples and comparisons and the first revised version was uploaded.
The final version will  be completed in few days.

The changes are marked by blue text.

---

### Public Comment · ~Anon_Yme1 · 2019-11-08
**General comment**

I think that the results presented are interesting because they illustrate clearly one important difficulty when dealing with learning of solutions for forward or backward problems with PDE. More precisely, the question is how to design learning-based PDE solvers that can ensure regularity of approximated solutions ? Usual cost functions make use of L2-norms that, as explained by the authors, can not make the difference between weak and strong approximated solutions. In fact, as it is also explained, measuring the accuracy of the approximation is a wide open problem. The recent result of Shen et al (2019) is a first step in this direction, but it is not completely satisfactory because the error is estimated with Lp norms that do not give any information about regularity.
The extra terms introduced in the cost functions are shown to be crucial to improve regularity of the approximations but the formal proof of this fact is surely not simple. In my opinion, the fact that this method avoids designing meshes is clearly a major advantage.

---

### Author Response · Authors · 2019-11-11
**General comments**

We uploaded a revised version of the manuscript.
Thanks to the constructive remarks of the reviewers, we considerably improved the text.
We highlighted our contribution, especially the promotion of strong solutions to the PDEs by our framework compared to recent approaches. We added comparisons of other inverse problems to Xu and Darve(2019).

We improved the notations, figure captions and discussion. Our modifications are marked in blue text.


Thank you very much!

---

### Decision · Program_Chairs · 2019-12-19

**Decision:**

Reject

**Comment:**

This paper proposes modifying the training loss for neural net-based PDE solvers, by adding an L_infty (max) term to the standard L_2 loss.  The motivation for this loss is sensible in that it matches the definition of a strong solution, but this is only a heuristic motivation, and is missing a theoretical analysis.

This paper's lack of novelty and polish, as well as the lack of clarity in the implementation details, makes this a narrow reject.